# Palm-Sized Wireless Transient Elastography System with Real-Time B-Mode Ultrasound Imaging Guidance: Toward Point-of-Care Liver Fibrosis Assessment

Zi-Hao Huang [1], Li-Ke Wang [1], Shang-Yu Cai [2], Hao-Xin Chen [2], Yongjin Zhou [2], Lok-Kan Cheng [1], Yi-Wei Lin [3], Ming-Hua Zheng [3,4] and Yong-Ping Zheng [1,5,*]

1. Department of Biomedical Engineering, The Hong Kong Polytechnic University, Hong Kong, China; 21068876r@connect.polyu.hk (Z.-H.H.); akewanglike@hotmail.com (L.-K.W.); lk-connie.cheng@polyu.edu.hk (L.-K.C.)
2. School of Biomedical Engineering, Medical School, Shenzhen University, Shenzhen 518000, China; 2110246012@email.szu.edu.cn (S.-Y.C.); chenhaoxin2021@email.szu.edu.cn (H.-X.C.); yjzhou@szu.edu.cn (Y.Z.)
3. MAFLD Research Center, Department of Hepatology, The First Affiliated Hospital of Wenzhou Medical University, Wenzhou 325000, China; lyw620558@163.com (Y.-W.L.); zhengmh@wmu.edu.cn (M.-H.Z.)
4. Key Laboratory of Diagnosis and Treatment for the Development of Chronic Liver Disease in Zhejiang Province, Wenzhou 325000, China
5. Research Institute for Smart Ageing, The Hong Kong Polytechnic University, Hong Kong, China
* Correspondence: yongping.zheng@polyu.edu.hk; Tel.: +852-2766-7664

**Abstract:** Transient elastography (TE), recommended by the WHO, is an established method for characterizing liver fibrosis via liver stiffness measurement (LSM). However, technical barriers remain towards point-of-care application, as conventional TE requires wired connections, possesses a bulky size, and lacks adequate imaging guidance for precise liver localization. In this work, we report the design, phantom validation, and clinical evaluation of a palm-sized TE system that enables simultaneous B-mode imaging and LSM. The performance of this system was validated experimentally using tissue-equivalent reference phantoms (1.45–75 kPa). Comparative studies against other liver elastography techniques, including conventional TE and two-dimensional shear wave elastography (2D-SWE), were performed to evaluate its reliability and validity in adults with various chronic liver diseases. Intra- and inter-operator reliability of LSM were established by an elastography expert and a novice. A good agreement was observed between the Young's modulus reported by the phantom manufacturer and this system (bias: 1.1–8.6%). Among 121 patients, liver stiffness measured by this system and conventional TE were highly correlated (r = 0.975) and strongly agreed with each other (mean difference: −0.77 kPa). Inter-correlation of this system with conventional TE and 2D-SWE was observed. Excellent-to-good operator reliability was demonstrated in 60 patients (ICCs: 0.824–0.913). We demonstrated the feasibility of employing a fully integrated phased array probe for reliable and valid LSM, guided by real-time B-mode imaging of liver anatomy. This system represents the first technical advancement toward point-of-care liver fibrosis assessment. Its small footprint, along with B-mode guidance capability, improves examination efficiency and scales up screening for liver fibrosis.

**Keywords:** transient elastography; liver stiffness measurement; liver fibrosis; liver elastography; point-of-care ultrasound; wireless ultrasound

## 1. Introduction

Chronic liver disease (CLD) represents a major and rising public healthcare issue, affecting an estimated 844 million individuals worldwide [1]. It can be caused by viral infections, such as hepatitis B or C virus, by excessive alcohol consumption, or by dietary habits leading to metabolic dysfunction-associated steatotic liver disease (MASLD), which

was formerly called non-alcoholic fatty liver disease (NAFLD). Liver fibrosis is the common pathway for CLDs of various etiologies, culminating in cirrhosis. Cirrhosis alone causes 1.16 million deaths globally, ranking as the 11th leading causes of death in 2015 [2]. Although early fibrosis has been shown to be partly reversible [3], cirrhosis is, by definition, deemed an irreversible condition. Hence, the goal of CLD management is to monitor liver fibrosis severity and prevent progression to cirrhosis.

Biopsy is historically the reference standard for diagnosing liver fibrosis. However, its invasive nature impedes long-term CLD monitoring. Histologic analysis of liver specimens is also subject to inter-pathologist variability and sampling error [4]. The serum biomarker method is minimally invasive, but its specificity and accuracy have been questioned [5]. These limitations have led to the rapid development of elastography modalities for assessing liver fibrosis in hepatology [6].

Shear wave-based elastography methods rely on shear wave propagation speed to quantify the mechanical properties of hepatic tissues, serving as a surrogate biomarker for liver fibrosis [7]. Although magnetic resonance elastography (MRE) has shown promising diagnostic accuracy against histology, its limited availability, high cost, and lengthy examination time have hindered its widespread clinical adoption. In contrast, the technological advantages of ultrasound elastography for characterizing the degree of liver fibrosis are being non-invasive, low-cost, rapid, of high patient acceptance, and having proven, excellent histopathological correlation. Two-dimensional shear wave elastography (2D-SWE) is one such technique that employs ultrasonically induced acoustic radiation force impulses to generate shear waves within the liver for liver stiffness measurement (LSM). 2D-SWE allows the simultaneous acquisition of anatomic B-mode images and stiffness mapping of the liver [8]. Another noteworthy technique is transient elastography (TE). TE by means of mechanical vibration excitation measures the speed of a 50 Hz shear wave propagating through the skin surface into the liver parenchyma [9]. Recommended by the World Health Organization (WHO) [10,11] and major clinical guidelines [6,7,12,13], TE stands as an established and best-validated approach for the first-line assessment of CLD. Numerous studies have validated LSM with TE for diagnosing liver fibrosis across various etiologies, and demonstrated its role in treatment follow-up and prognosis prediction [7,12].

Despite its success in hepatology, a commonly reported limitation of TE is the lack of sufficient visual guidance for sampling the liver of interest [12–17]. Conventional TE has primarily focused on the design of a single-element transducer, providing LSM alongside M-mode. Because no anatomic B-mode images are provided, TE practitioners have difficulty in targeting a liver parenchymal region devoid of blood vessels or focal lesions for analysis. This limitation contributes to the increasing frequency of TE failure or unreliable results and significantly reduces examination efficiency. Additionally, conventional TE systems have limited clinical acceptance due to their bulky, immobile, and wired natures [18]. The existing setup confines TE assessment to an examination room, potentially limiting its utility beyond a hospital setting. For example, due to the rising prevalence of CLDs and associated risk factors such as obesity, Pere et al. [19] advocated for programs of screening for liver fibrosis. Equipment size is a practical consideration for implementing large-scale liver examinations in the community [20]. This becomes another room for improving TE techniques and expanding its clinical applications. In light of these obstacles, there is a need for incorporating the capability of B-mode imaging and point-of-care ultrasound (POCUS) into the TE examination procedure to make TE suitable for point-of-care applications.

In this work, we introduce a newly developed TE system (named 'Liverscan') which is palm-sized in dimension and enables LSM guided by real-time B-mode imaging of liver morphology. Despite its potential utility beyond routine clinical practice, the Liverscan system has yet to undergo clinical scrutiny. Hence, the objectives of this methodological cross-sectional study were to assess the accuracy of LSM using Liverscan on tissue-mimicking phantoms, and the reliability and validity of LSM in adult patients with confirmed CLD of various etiologies.

## 2. Materials and Methods

### 2.1. Introduction to the Liverscan System

The novel Liverscan® system (Eieling Technology Limited, Hong Kong, China) comprises an all-in-one probe and a paired software installed in a terminal device (Figure 1). The battery-driven probe wirelessly connects to a Windows-based notebook computer or tablet via Wi-Fi communication, and houses a voice-coil motor mounted on the axis of a 3.5 MHz phased array ultrasonic transducer. This engineering design allows simultaneous B-mode imaging and stiffness measurement of the liver with a single integrated probe. The cylinder-like probe measures 21.1 cm in length, 5.1 cm in width, and 6.4 cm in height. Its tip serves as both a low-frequency vibrator to generate shear waves and an ultrasonic transmitter–receiver to trace wave propagation. Specifically, the vibrator induces shear waves through external mechanical vibration with low frequency (50 Hz) and mild amplitude. The phased array transducer, configured with 32 elements operating at 20 frames/s, produces real-time B-mode images in a sector field of view (FOV). A single beam centred at the array acquires RF signals at an ultra-fast rate of 6400 frames/s and within a short time span of 80 ms to image the axial component of the shear wave along the beam direction. RF data acquisition from this single scan line synchronizes with the start of mechanical excitation. The system allows for the adjustment of the region of interest (ROI) location axially and radially, aiding in the avoidance of reverberation artifacts beneath Glisson's capsule [13,21]. In this study, the ROI is set at a depth range of 25–65 mm, consistent with other clinically available TE systems.

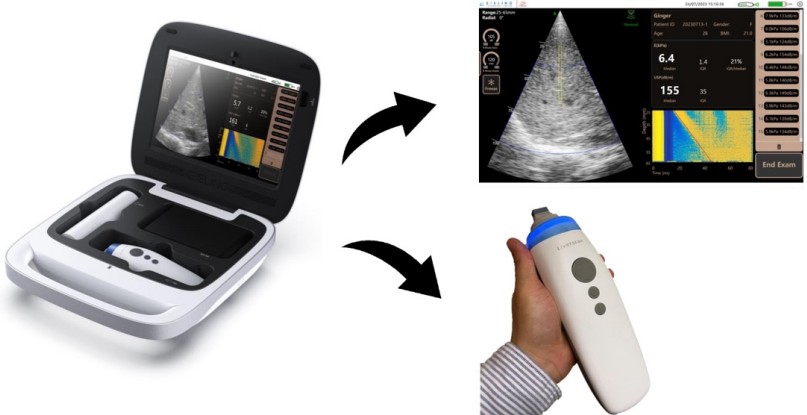

**Figure 1.** Schematic diagram of the Liverscan probe and paired software interface. Working principle of the Liverscan system: LSM procedure including shear wave generation and ultrasound acquisition is guided by real-time B-mode imaging. The displacements of the elastic liver tissue caused by the passage of the shear wave are documented via ultra-fast RF acquisitions, resulting in the formation of an elastogram. The elastogram, which estimates the Young's modulus of the liver, is visualized as a strain image as a function of time and depth.

To construct an elastogram (a shear wave propagation map), axial displacements are estimated from the sequences of RF signals using a texture disturbance detection approach. The elastogram appears as a 2D color-coded image representing the spatial–temporal information of the propagating shear wave. A MobileNet-based regression algorithm detects and computes the shear wave trajectory, reflecting liver tissue deformation associated with the passage of the shear wave. Shear wave speed is deduced from the slope of the trajectory pattern and converted into the Young's modulus using the equation:

$$E = 3\rho V_s^2 \tag{1}$$

where $E$ is the Young's modulus, $V_s$ is the shear wave speed, and $\rho$ is the assumed tissue density (1000 kg/m$^3$, equivalent to water density).

A computer terminal is used to display B-mode images in real-time alongside the algorithmic calculation of the Young's modulus and corresponding elastogram. This stand-alone processing unit contributes to the downsizing of the probe.

### 2.2. Study Design

To evaluate the accuracy of Liverscan in quantifying the Young's modulus, a series of phantoms with established reference values were used (Figure 2). Between May 2022 and June 2023, a large clinical trial was conducted, and adult subjects were recruited from two participating institutions: The Hong Kong Polytechnic University and The First Affiliated Hospital of Wenzhou Medical University. To evaluate the feasibility and validity of Liverscan in a clinical setting, we performed head-to-head comparisons of its LSM with other clinically available ultrasound elastography systems: Fibroscan® (fs, Echosens, Paris, France) and Fibrotouch® (ft, Hisky, Wuxi, China), which are based on conventional TE, and Aixplorer® (SuperSonic Imagine, Aix-en-Provence, France), based on 2D-SWE (Figure 2). We employed different statistical analyses and compared the results to investigate the variability among the elastography modalities. Additionally, the intra- and inter-operator reliability of LSM using Liverscan were established.

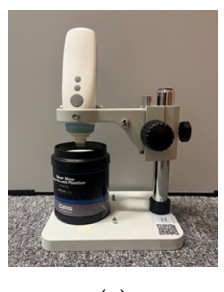

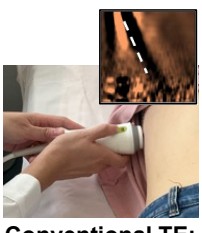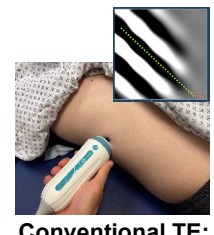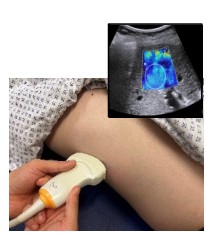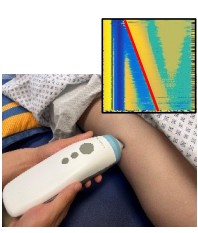

**Conventional TE:** Fibroscan®    **Conventional TE:** Fibrotouch®    **2D-SWE:** Aixplorer®    **B-mode guided TE:** Liverscan®

(**a**)    (**b**)

**Figure 2.** Experimental setup for Liverscan validation. (**a**) Phantom experiment setup: the probe was placed over the phantom using a custom testing platform; (**b**) Human experiment setup for the comparative study: four liver elastography techniques in a randomized order were administrated to each subject on the same day.

In the Hong Kong cohort, the examinations of Liverscan, conventional TE-ft, and 2D-SWE were conducted on the same day of the examination of conventional TE-fs. While subjects received conventional TE-fs at an external health check clinic (Virtus Medical Centre, Hong Kong, China) where a Fibroscan system is installed, other elastography examinations were conducted at the Clinical Ultrasound Lab of The Hong Kong Polytechnic University. The examination order among different liver elastography techniques was randomized for each subject. In the Wenzhou cohort, patients underwent the conventional TE-fs and Liverscan examinations on the same day. Demographic, anthropometric, and clinical data were recorded by research coordinators using standard protocols. All operators participating in this study were blinded to the patients' clinical and other liver elastography data. Ethical approval was obtained from the Institutional Review Board of The Hong Kong Polytechnic University (HSEARS20210809002). This prospective, cross-sectional, dual-center imaging study complies with the Declarations of Helsinki and the STROBE reporting criteria.

### 2.3. Phantoms

In order to quantitatively study the accuracy of Liverscan in estimating the Young's modulus, experiments were carried out on multiple commercial elasticity phantoms. As part of Model 039 (Computerized Imaging Reference Systems (CIRS), Inc., Norfolk, VA, USA) and KS215 T-3 (Institute of Acoustics, Chinese Academy of Science, Beijing, China), these phantoms were specifically designed for quality control and performance testing of clinical ultrasound-based elastography systems. To eliminate the possible impact of viscosity and ease direct

comparison among elastography techniques, each phantom contains homogeneous, isotropic, and nearly completely elastic materials. In this study, eight phantoms with respective Young's modulus values of 1.2 kPa, 2 kPa, 8 kPa, 12 kPa, 18 kPa, and 72 kPa were tested. These nominal values encompass the elastic properties spanning healthy through cirrhotic livers and serve as the ground-truth for assessing measurement accuracy.

To obtain measurements, the Liverscan probe was mounted on top of the phantoms and fixed using a probe holder. Ultrasound gel was applied to the phantom surface to ensure acoustic coupling. Twenty valid measurements were carried out in the different locations of each phantom, with the tip of the probe in contact with the phantom surface to maintain mechanical coupling. A static applied force between the transducer and phantom surface was monitored and standardized at 5 N across the phantoms. A single operator, blinded to the phantom being tested and its LSM result, was involved. The order of data acquisition was randomized for phantoms. Each stiffness value represents the Young's modulus of the tested phantom, and the mean and median of the twenty phantom measurements were calculated for subsequent statistical analyses.

### 2.4. Subjects

A *priori* analysis using the G*Power software (Heinrich-Heine-Universität Düsseldorf, Düsseldorf, Germany) projected a sample size of 84 subjects for the Pearson correlation, with a medium Cohen's effect size $r = 0.3$, statistical power of 0.8, and a two-sided significance level of 0.05 under a bivariate correlation design. The final target sample size was set at 121 to allow for 30% dropout [22]. Adult subjects were enrolled if they had been diagnosed with CLD caused by hepatitis B virus (HBV), hepatitis C virus (HCV), MASLD, alcoholic liver disease (ALD), or autoimmune hepatitis (AIH) and clinically indicated for follow-up of CLD. Subjects were excluded in the case of ascites, pregnancy, active implantable cardiac device, liver transplantation, and refusal to receive contemporaneous liver elastography examinations. All subjects gave written informed consent and fasted for a minimum of 3 h before an examination procedure.

A total of 121 patients were divided into three subgroups used for different statistical analyses. Specifically, LSM by Liverscan was validated against that by conventional TE-fs in the complete cohort of 121 subjects, in a dual-center setting. We further evaluated the association of Liverscan, conventional TE-ft, and 2D-SWE, in comparison with conventional TE-fs, in the subgroup of 90 subjects. In addition to the comparative studies on the validity of Liverscan, subjects were randomly invited to participate in a sub-analysis assessing operator reliability. Given the expected success rate with Liverscan set at 95%, a projected sample size of 60 was deemed necessary to estimate an excellent intraclass correlation coefficient (ICC) of $0.9 \pm 0.05$ and a dropout rate of 5% under a two-repetition design [23]. The reliability analysis involved two operators with different levels of experience and expertise; Operator 1 with a background in radiology had six years of experience in TE development and operation, while Operator 2 was a novice trained as an engineer.

### 2.5. TE Examination

Based on the technical principle, TE examinations performed in this study fall into two categories: (**a**) **conventional TE** without B-mode guidance and (**b**) **B-mode guided TE**.

In the conventional TE session, the Fibroscan (fs) and Fibrotouch (ft) systems were used. According to the procedure described previously [6,7,9,12,13], subjects were asked to lie down in the dorsal decubitus position with the right arm in maximal abduction and placed behind their heads. The transducer tip of the probe was placed onto the skin surface between the ribs at the level of the right lobe of the liver. The Fibroscan XL probe was administrated to obese patients as recommended by an automatic probe selection tool [24]. Subjects who received the examination of conventional TE-fs using either the standard M or XL probe were included in this study. The examination of conventional TE-ft was performed with one universal probe designed for varying patient morphotypes.

A minimum of 10 valid acquisitions with conventional TE-fs and TE-ft were performed on each subject, respectively.

In the B-mode guided TE session using Liverscan, the same examination procedure, including patient positioning and measurement protocol, were applied as outlined above. Assisted by real-time B-mode, a sufficiently thick portion of the liver parenchyma free of large vascular structures was identified and consecutively measured ten times. In the intra- and inter-operator reliability study, subjects were assessed twice by each of two operators, generating four experimental conditions per subject:

- First examination of Operator 1;
- Second examination of Operator 1;
- First examination of Operator 2;
- Second examination of Operator 2.

To avoid bias by a learning effect, Operator 1 and 2 independently performed the Liverscan examinations in a randomized order, each operator being blinded to the LSM results of the other. During the 2nd examination procedure, the operator was blinded to the previously obtained LSM result of the 1st examination. The Liverscan probe was removed from the patient and repositioned between conditions, with the interval no less than 10 min.

Overall, the same statistical procedure was applied to derive liver stiffness from the three TE examinations with conventional TE-fs, conventional TE-ft, and B-mode guided TE. The median stiffness value (in kPa) of the 10 valid acquisitions was considered representative of the Young's modulus of the liver and was calculated for each of the three TE examinations, respectively. As an indicator of variability between LSMs, the ratio of the interquartile range to the median value (IQR/median) was calculated. TE results were included in the final analysis if at least 10 valid acquisitions were made with an IQR/median $\geq$30% and a success rate $\geq$60%.

### 2.6. SWE Examination

2D-SWE was performed using a convex probe (XC6–1, central frequency: 3.5 MHz) of the Aixplorer® system (SuperSonic Imagine, Aix-en-Provence, France) via the right intercostal approach [8,21,25]. Subjects were in the supine position with the right arm in maximal abduction. During the stiffness acquisition process, a rectangular SWE box was placed in the right liver parenchyma free of blood vessels, portal tracts, and focal lesions, at least 1 cm in depth from Glisson's capsule and between 2 and 7 cm from the skin surface. Ten frames of SWE images were obtained from the right hepatic lobe during repeated inspiratory breath-holds. In the subsequent data analysis, a 1 cm diameter circular ROI was placed retrospectively over a region of a relatively homogeneous portion at a depth of 3–6 cm of the SWE box for LSM. To average the possible effect of heterogenous fibrosis, a third-party rater placed five individual ROIs in different sub-regions of each SWE box. The average of five SWE values derived from the circular ROIs and expressed in kPa was kept as the representative result for that SWE image. In this fashion, each SWE image was measured five times producing a total of 50 SWE value acquisitions for each subject. The median of the 50 SWE values obtained from the right-lobe segments was calculated and used in this study as the per-patient Young's modulus of the liver.

### 2.7. Statistical Analyses

All statistical analyses were performed using Statistical Package for Social Sciences (SPSS, IBM corporation, Armonk, NY, USA). The level of significance was set at 0.05.

**Descriptive statistics.** Cohort characteristics were descriptively summarized, expressing continuous variables as a median (interquartile range [IQR]) and categorical variables as absolute figures with percentages. Confidence intervals (CIs) were reported at the 95% level. A Friedman two-way ANOVA with post-hoc Dunn's test was conducted to compare the liver stiffness (kPa) and its corresponding IQR/median of LSM (%) across four liver elastography techniques, respectively.

**Accuracy statistics.** Comparisons were made between the Young's modulus of the phantoms measured by Liverscan and that reported by the manufacturers. According to the Radiological Society of North America Quantitative Imaging Biomarker Alliance (QIBA) [26], three technical performance metrics of linearity, bias, and precision were used for the performance assessment of phantom Young's modulus quantification. Linearity and bias together assess the degree to which LSM provides an estimate of the true value. Linearity was analyzed using a first-order polynomial regression model of the average of measured values against reference values. Bias, defined as the percentage error between the two groups of values, was calculated for each phantom using the formula:

$$\text{Bias (\%)} = \frac{|Liverscan\ value\ (\text{kPa}) -\ Reference\ value\ (\text{kPa})|}{Reference\ value\ (\text{kPa})} \times 100\% \qquad (2)$$

Precision assesses the relative variability between twenty repeated measurements, and was analyzed using the coefficient of variation (CV).

**Validity statistics.** Correlation relationships were examined among liver stiffness values measured by conventional TE-fs, conventional TE-ft, Liverscan, and 2D-SWE, respectively. A correlation matrix was constructed to visualize the pairwise comparison of Pearson correlation coefficients, whereby $0.1 < |r| < 0.3$ was considered weak correlation, $0.3 < |r| < 0.5$ moderate correlation, and $|r| > 0.5$ strong correlation. Using convectional TE-fs as the benchmark technique, Bland–Altman plots were constructed to examine the agreement of each pair. The linear relationship of convectional TE-fs against the other three elastography techniques was evaluated pairwise using the simple linear regression model. The Pearson correlation coefficient ($r$) as the strength metric of association, coefficient of determination ($R^2$) as the strength metric of linearity, and mean difference between techniques as the strength metric of agreement were reported for each pair.

**Reliability statistics.** We used intraclass correlation coefficient (ICC), standard error of measurement (SEM), and coefficient of variation method error ($CV_{ME}$) to assess the inter-operator and intra-operator agreement for stiffness values derived from Liverscan. ICC was interpreted as follows: 0.90–1.00 = excellent, 0.75–0.90 = good, 0.5–0.75 = fair, and <0.5 = poor, according to Koo & Li's conventional criteria [27].

## 3. Results

**Phantom measurement accuracy.** As presented in Table 1, the phantom results demonstrated that the measured values (both mean and median) for all phantoms were very close to or within the allowed error range provided by the phantom manufacturers. The linearity across eight reference phantoms is plotted in Figure 3, with a first-order linear model indicating very strong linear fit ($R^2 = 1$). The precision, as assessed by the CVs of twenty repeated measurements, ranged from 1.0–15.5%, and bias was less than 10% for all phantoms.

**Table 1.** Accuracy analysis of phantom Young's modulus measurement across eight phantoms.

|  | Ground-Truth (kPa) | Mean ± SD/Median [IQR] (kPa) * | Bias (%) | CV (%) |
|---|---|---|---|---|
| Phantom #1 | 1.45 ± 0.1 | 1.6 ± 0.2/1.5 [0.3] | 8.6 | 15.5 |
| Phantom #2 | 2.22 ± 0.2 | 2.2 ± 0.0/2.2 [0.0] | 1.1 | 1.0 |
| Phantom #3 | 4.66 ± 0.5 | 4.7 ± 0.2/4.7 [0.2] | 1.4 | 3.6 |
| Phantom #4 | 7.0 ± 0.2 | 6.9 ± 0.3/7.0 [0.2] | 1.9 | 4.7 |
| Phantom #5 | 12.8 ± 1.3 | 13.1 ± 0.5/13.1 [0.6] | 2.6 | 3.7 |
| Phantom #6 | 16.2 ± 0.6 | 16.6 ± 0.5/16.6 [0.3] | 2.2 | 2.8 |
| Phantom #7 | 32.1 ± 0.4 | 32.9 ± 1.9/33.0 [2.7] | 2.5 | 5.8 |
| Phantom #8 | 76.9 ± 7.7 | 78.8 ± 4.5/77.9 [6.3] | 2.5 | 5.7 |

IQR = interquartile range; SD = standard deviation; CV = coefficient of variation. * The Young's modulus of the phantoms was expressed as mean ± SD or median [IQR] for twenty repeated measurements.

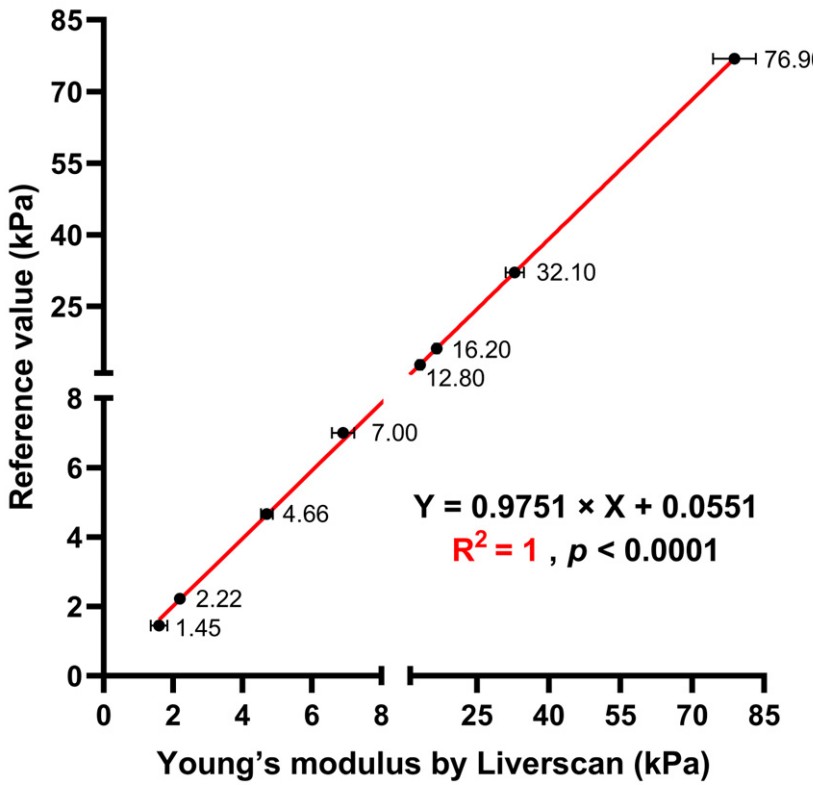

**Figure 3.** Measurement of Young's modulus across eight phantoms with known reference values: scatterplot showing the linear relationship of the Young's modulus of the phantoms derived by Liverscan against the truth values.

**Patient characteristics.** Between May 2022 and June 2023, a convenience sampling approach was adopted to recruit 121 eligible subjects with a history of known CLD for this study. The characteristics of the 121 patients are outlined in Table 2. Ninety patients were recruited at The Hong Kong Polytechnic University and thirty-one at The First Affiliated Hospital of Wenzhou Medical University. Fifty-four were women (45%) and the median age was 54 years (range: 22–73 with an IQR of 18). The majority (41%) of patients were experiencing CHB (additional 21% with coexistent steatosis) or CHC, and 35% had MASLD. The median BMI was 24 kg/m$^2$ (range: 16–41 with an IQR of 5). Over one-third of the patients (35%) were overweight or obese, and more than half (57%) were classified as having central obesity. The median skin–capsular distance was 13 mm (range: 5–26 with an IQR of 7).

**Inter-technique difference comparison.** Figure 4 summarizes the aggregated LSMs and IQR/median results from the four-technique comparisons in the subgroup of 90 patients. A Friedman ANOVA identified differences in median LSMs among conventional TE-fs (4.6 kPa, IQR: 3.7–5.8), conventional TE-ft (5.8 kPa, IQR: 4.6–6.9), Liverscan (5.3 kPa, IQR: 4.5–6.4), and 2D-SWE (5.8 kPa, IQR: 4.7–6.7), $\chi^2(3) = 64.366$, $p < 0.001$. Post-hoc Dunn's test revealed that Liverscan and 2D-SWE produced the similar stiffness values ($p = 0.364$), which were significantly greater than those by conventional TE-fs (all $p < 0.001$). The aggregated variability of LSMs, as assessed by the ratio of IQR/median, significantly differed across the four techniques, except for conventional TE-fs and 2D-SWE ($p = 1.000$). Notably, Liverscan exhibited the significantly highest median IQR/median value (23% with an IQR of 18–31%, $p < 0.001$), compared with conventional TE-fs (12% with an IQR of 7–15%), conventional TE-ft (8% with an IQR of 4–11%), and 2D-SWE (12% with an IQR of 8–18%).

**Table 2.** Characteristics of the study cohort (n = 121).

| Characteristics | Median [IQR] or Proportion% (n) |
|---|---|
| • Demographics | |
| Male | 55% (67) |
| Age, years | 54 [18; range 22–73] |
| Overweight & obese | 35% (42) |
| Central obesity | 57% (69) |
| Metabolic syndrome | 15% (18) |
| • Anthropometrics | |
| Weight, kg | 65 [16; range 43–119] |
| BMI, kg/m$^2$ | 24 [5; range 16–41] |
| Waist circumference, cm | 84 [17; range 65–131] |
| Skin-to-liver capsule distance, mm | 13 [7, range 5–26] |
| • Liver disease etiology | |
| Viral (CHB, CHC) | 41% (50) |
| MASLD | 35% (42) |
| ALD | 3% (4) |
| Coexistence of HBV and MASLD | 21% (25) |
| • Liver stiffness, kPa | |
| Conventional TE (Fibroscan) | 5.4 [4.5; range 2.4–39.9] |
| B-mode guided TE (Liverscan) | 5.9 [5.4; range 2.7–38.2] |
| • IQR/median of LSM, % | |
| Conventional TE (Fibroscan) | 12 [8; range 3–27] |
| B-mode guided TE (Liverscan) | 23 [13; range 8–45] |

IQR = interquartile range; BMI = body mass index; LSM = liver stiffness measurement; MASLD = metabolic dysfunction-associated steatotic liver disease; CHB = chronic hepatitis B; CHC = chronic hepatitis C; ALD = alcoholic liver disease.

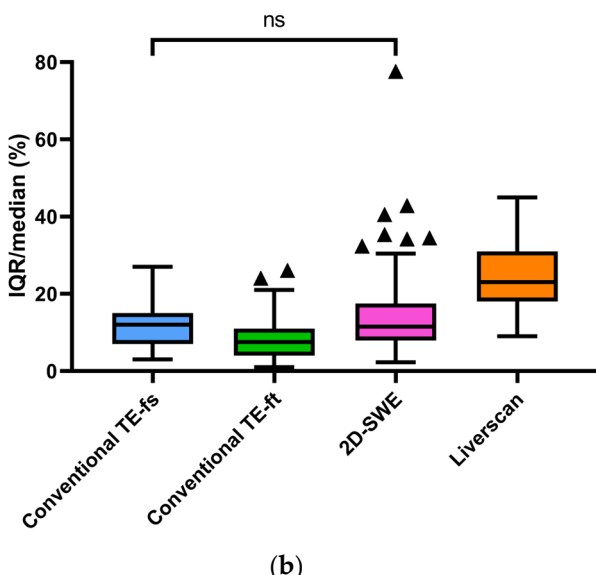

**Figure 4.** Aggregated results of (**a**) liver stiffness (kPa) and (**b**) IQR/median of LSMs (%) among four elastography techniques (ns: $p > 0.05$; $p \leq 0.05$ otherwise).

**Operator reliability.** The intra-operator and inter-operator reproducibility of LSM using Liverscan was analyzed in a subgroup of 60 patients (Table 3). For intra-operator reliability, ICC$_{(3,1)}$ was 0.913 (95% CI: 0.859 to 0.947) for Operator 1 and 0.890 (95% CI: 0.822 to 0.933) for Operator 2. To examine whether inter-operator reliability varied with operation experience, we grouped two operators for each of the first and second examination sessions and calculated two independent sets of inter-operator ICCs for comparison. Results showed that the inter-operator

$ICC_{(2,1)}$ for the second examination (0.856, 95% CI: 0.770 to 0.911) was marginally larger than that for the first examination (0.824, 95% CI: 0.721 to 0.891). The SEM and $CV_{ME}$ of all measurements ranged from 0.6 to 0.8 kPa and from 9.6 to 14.0%, respectively.

**Table 3.** Intra- and inter-operator reliability results for liver stiffness measurement (n = 60).

| Liver Stiffness (kPa) | $ICC_{(3,1)}$ [a]/$_{(2,1)}$ [b] (95% CI) | SEM (kPa) | $CV_{ME}$ (%) |
|---|---|---|---|
| **Intra-operator reliability:** | | | |
| Operator 1 | 0.913 (0.859–0.947) [a] | 0.558 | 9.605 |
| Operator 2 | 0.890 (0.822–0.933) [a] | 0.679 | 11.727 |
| **Inter-operator reliability:** | | | |
| Examination 1 by Operator 1 vs. 2 | 0.824 (0.721–0.891) [b] | 0.814 | 14.011 |
| Examination 2 by Operator 1 vs. 2 | 0.856 (0.770–0.911) [b] | 0.761 | 13.143 |

ICC = intraclass correlation coefficient; CI = confidence interval; SEM = standard error of measurement; $CV_{ME}$ = coefficient of variation and method error. Intra-operator ICC: tests between two repeated examinations conducted by Operator 1 and 2, respectively. Inter-operator ICC of Examination 1: tests between the 1st examination conducted by Operator 1 vs. the 1st examination conducted by Operator 2. Inter-operator ICC of Examination 2: tests between the 2nd examination conducted by Operator 1 vs. the 2nd examination conducted by Operator 2. [a] ICC computed using two-way mixed model and consistency; [b] ICC computed using two-way random model and consistency.

**Pairwise validity comparison.** Of the 121 patients, liver stiffness as assessed by conventional TE-fs and Liverscan was highly correlated ($r = 0.975$, 95% CI: 0.964 to 0.982, $p < 0.001$). The relationship between liver stiffness for each subject was fitted to a linear regression model ($Y = 0.9544 \times X + 1.107$) and the linearity was strong ($R^2 = 0.950$). A Bland–Altman plot showed a small mean difference of −0.77 kPa (95% limit of agreement: −3.44 to 1.90) between techniques, indicating minimal overestimation by using Liverscan (Figure 5).

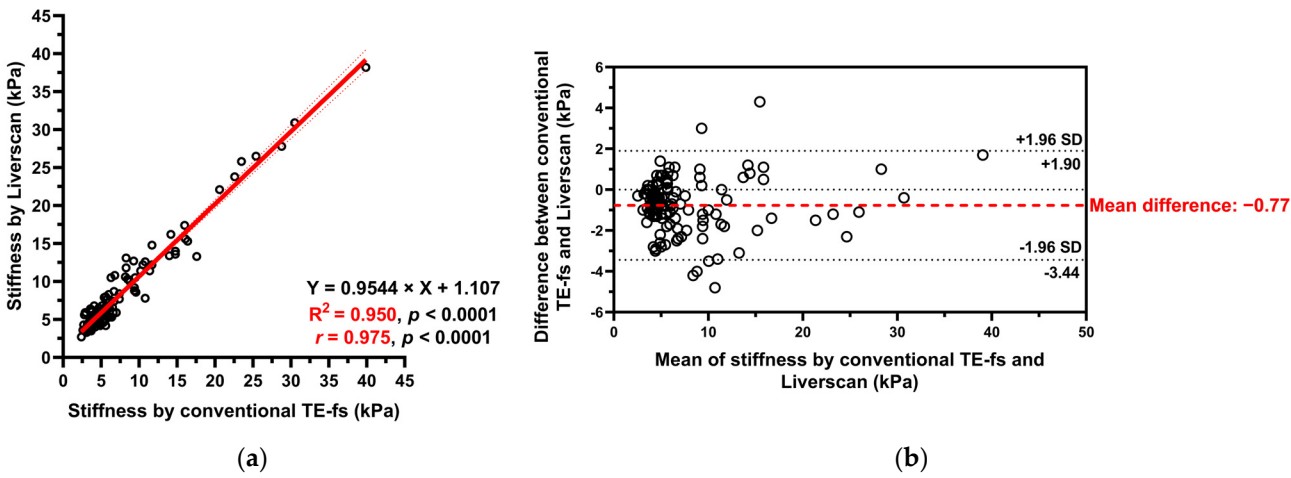

**Figure 5.** Liver stiffness measured by conventional TE-fs vs. Liverscan. (**a**) Scatterplot with Pearson correlation and simple linear regression analysis; (**b**) Bland–Altman analysis.

The correlation matrix results of 90 patients showed positive and statistically significant Pearson correlation coefficients of all pairs ($r$ ranging from 0.55 to 0.79, $p < 0.001$), indicating that liver stiffness measured by the four techniques were inter-correlated (Figure 6). As the only technique allowing the mapping of fibrotic distribution, 2D-SWE exhibited a similar correlation with conventional TE-fs ($r = 0.62$, 95% CI: 0.471 to 0.731, $p < 0.001$) and Liverscan ($r = 0.66$, 95% CI: 0.524 to 0.762, $p < 0.001$). The strongest correlation was observed in the pair of conventional TE-fs vs. Liverscan, with a Pearson correlation coefficient of 0.79 (95% CI: 0.691 to 0.854, $p < 0.001$). Conventional TE-fs correlated nominally more with Liverscan ($r = 0.79$) than with conventional TE-ft ($r = 0.65$, 95% CI: 0.507 to 0.752, $p < 0.001$) and with 2D-SWE ($r = 0.62$, 95% CI: 0.471 to 0.731, $p < 0.001$). This relationship was confirmed by linear regression and Bland–Altman analyses, where conventional TE-fs

and Liverscan demonstrated the lower mean difference of −0.69 kPa (95% limit of agreement: −2.91 to 1.54) and stronger linearity ($R^2$ = 0.617, $p < 0.001$) than other pairs (See Supplementary Figures S1 and S2).

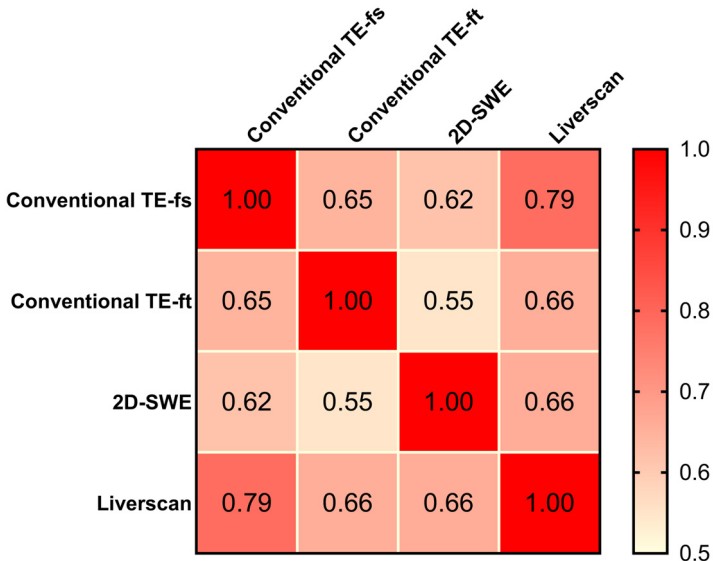

**Figure 6.** Pearson correlation matrix: comparison among four elastography techniques.

## 4. Discussion

The role of TE in hepatology is well-established and unquestionable; however, challenges persist regarding its accessibility and utility. For example, examination inefficiency and failure related to inadequate imaging guidance is not uncommon during a LSM procedure. The large size and immobility of conventional TE systems also limits their use in contexts where inclusion could optimize patient care workflow, such as ward rounds and office consultations, i.e., point-of-care applications. Addressing these issues requires engineering efforts to further advance the TE examination procedure. This is our motivation to develop a palm-sized wireless TE system with real-time B-mode as a guidance tool, aiming to maximize the efficacy of TE in both inpatient and outpatient settings. This study adopted a head-to-head comparison design, supporting the feasibility and validity of Liverscan as a liver elastography modality compared to three other existing systems. In particular, its performance of LSM was demonstrated by small biases (range 1.1–8.6%) for eight reference phantoms and a high correlation ($r$ = 0.975) between liver stiffness measured by conventional TE-fs and Liverscan in 121 patients with varying degrees of fibrosis severity. Moreover, LSM by Liverscan provided high intra- and inter-operator agreement (ICC range 0.824–0.913).

**Phantom analysis.** The first step toward validating the performance of Liverscan was to perform a study on calibrated phantoms devoid of potential *in vivo* confounders, such as respiratory motion and liver heterogeneity. Experiments involving eight phantoms showed that this system can accurately estimate the Young's modulus. Discrepancies from the known reference values were considered acceptable, with a bias of less than 10% in all phantoms and a mean bias of 2.9 ± 2.4%. Of note, small biases of 1.9–2.6% were observed for phantoms within the clinically relevant range of 7 kPa to 16 kPa, representing progressive fibrosis stages. The precision of twenty measurements per phantom proved satisfactory, with the CVs below 16%. The small CVs reflected the homogeneity of the phantom materials, as repeated measurements were taken at different locations. This finding aligned with prior results from two other phantom studies, which had reported CVs of 0.6–9.8% [28] for 2D-SWE and 0–9% for TE [29].

**Operator reliability.** Our results affirmed that LSM by Liverscan was highly reproducible, characterized by excellent-to-good inter- and intra-operator agreement. The

reported ICCs between 0.824 and 0.913 were similar to those shown in one reliability study applying conventional TE to a CHB cohort (ICCs: 0.86–0.96) [30]. The reproducibility of Liverscan was even higher than another large MASLD study [31], with the intra-operator ICC of 0.837 and inter-operator ICC of 0.790 specifically reported for conventional TE and the intra-operator ICC of 0.847 and inter-operator ICC of 0.705 specifically reported for 2D-SWE. Nevertheless, caution is warranted in making such direct comparisons, because the patient samples involved were not exactly the same. Excellent-to-good intra-operator reliability ($ICC_{(3,1)}$ of nearly 0.9 observed in both the experienced and novice operators) suggested that medical background and expertise may not be a necessity for obtaining reproducible results. Additionally, inter-operator reliability analyses confirmed good agreement in two independent sessions, but inferior to the intra-operator reliability. Inter-operator $ICC_{(2,1)}$ of 0.856 for the second examination appeared slightly higher than that of 0.824 for the first examination, with a minimum 10 min interval between them. Similar trends have been reported by Fang et al. [32], who noted the improved inter-operator reliability for 2D-SWE from the previous to latter session, separated by two weeks. We attributed this improvement to growing operational experience or learning retention. However, it remains debated whether the ICC increase of 0.032 represents a clinically meaningful change. Further validation is warranted in a larger cohort and among new operators with varying training levels. The range of the $CV_{ME}$ values (9.6–14.0%) suggested an acceptable degree of variability, with less variable measurements observed in intra-operator settings. Although no SEM data exist on the topic of liver elastography for comparison, the SEM values in our study (0.6–0.8 kPa) were deemed quite low. These SEM values were lower than those reported in a previous elastography study on lateral abdominal muscles (SEM: 7.8–10.7 kPa) [33]. Nevertheless, it is pertinent to excise caution in interpreting reliability results because biologic factors (rather than technical factors) may contribute to the variability in LSMs, especially in individuals with nonuniform distribution of liver fibrosis. On the other hand, the absence of overlying subcutaneous tissues that attenuate waves and homogeneous phantom materials may explain the less variable results observed in phantoms compared with human livers.

**Validity comparison.** This prospective study supported that B-mode guided TE can benchmark with conventional TE with respect to LSM. The Bland–Altman plot of Liverscan relative to conventional TE-fs as the reference technique revealed a minimal mean difference (−0.77 kPa) and limits of agreement within ±4 kPa. A strong correlation and linearity between liver stiffness measured using both techniques ($r = 0.975$, $R^2 = 0.950$) were also established in a CLD cohort diagnosed with ALD, MASLD, CHB, and CHC. We further examined the association of contemporaneous use of different liver elastography techniques. The correlation between conventional TE-fs and 2D-SWE ($r = 0.618$) was consistent with that of previous literature [34,35], which in part supported the validity of other pairwise comparisons analyzed in this study. Inter-correlation among the four techniques was observed in 90 CLD patients, with the strongest correlation ($r = 0.785$), strongest linear relationship ($R^2 = 0.617$), and highest inter-technique agreement (mean difference = −0.69) found at the pair of conventional TE-fs vs. Liverscan. These findings demonstrated that Liverscan was superior to conventional TE-ft and 2D-SWE when validated against conventional TE-fs.

Previous literature suggested shear wave frequency-dependent discrepancies in LSM between TE and 2D-SWE [18,34]. This could be explained by the lower frequency used by TE (50 Hz) compared to the broadband of 60 to 600 Hz used by 2D-SWE [8,36]. Interestingly, our study did not find such a discrepancy to be the case, because 2D-SWE and Liverscan had the tendency to produce not significantly different aggregated LSMs ($p = 0.364$). This probably relates to their technical similarity that allows LSM under B-mode guidance to avoid interference from other non-liver tissues. Furthermore, previous diagnostic studies [8,18,21] suggested that 2D-SWE measurement compared favorably to conventional TE, with a generally higher area under the curve (AUC) in differentiating

each fibrosis stage. Future research should compare the diagnostic accuracy of 2D-SWE and B-mode guided TE against histology.

On the other hand, LSMs by Liverscan appeared to be less consistent among individual measurements, as reflected by higher IQR/median values (median: 23%, range: 9–45%). According to the commonly defined reliability criteria and standard practice of using TE in clinical settings, an IQR/median smaller than 30% [37,38] is necessary for valid and high-quality measurement results. Therefore, the relatively high IQR/median values of Liverscan demonstrated in this study would not affect the measurement, as they nearly align with the acceptable reliability category. One potential reason for Liverscan yielding a relatively larger IQR/median would be the relatively smaller beam width currently used in the device. In the current hardware, we adopted the same beamforming parameters for both TE mode and B-mode imaging, resulting in a beam width of approximately 2.3 mm at the focal zone. This specification is smaller than the 10 mm reported by conventional TE [12] and the ROI width of 10 mm [25] standardized for 2D-SWE in this study. Considering the heterogeneity in hepatic fibrosis, a smaller beam width could lead to greater variation among different measurements, as the liver keeps moving. In future studies, we plan to increase the beam width by adjusting the beamforming parameters to investigate whether the IQR/median value will be reduced accordingly. Meanwhile, maintaining a smaller beam width may provide us with a unique opportunity to map the Young's modulus distribution for assessing liver fibrosis heterogeneity, by steering the beam towards different locations while conducting TE measurements. Our future study will also investigate this possibility.

**Highlight of small footprint.** Given the prevalence of fibrosis $\geq$F2 of 27% in high-risk populations [39] and 7.5% in the general population [40], it is imperative for healthcare providers to have a technique that is not only accurate and reproducible, but also highly accessible and cost-effective for assessing liver fibrosis at the point of care. The alarmingly high prevalence further strengthens the necessity of preventive screening programs for liver fibrosis [19]. However, size constraint has been a practical limitation in the widespread availability of TE systems and the scale-up of liver screening programs, which motivated us to introduce POCUS into a TE examination procedure. Until this work, no engineering attempt had been made to downsize an ultrasound elastography system into a palm-sized version. Another key innovation is that Liverscan is the first wireless ultrasound elastography system; its small footprint represents only a fraction of the size of conventional ones. This portability opens up new possibilities for conducting home-based primary care in a non-hospital setting and screening on a larger scale. Its cable-free connection and lightweight design greatly enhance operational flexibility. Its high accessibility and ease of use make TE a community-based risk-stratification strategy, allowing for the earlier identification of patients and potentially leading to cost savings for healthcare systems. We believe this new point-of-care testing method could change the paradigm of how the liver fibrosis epidemic is addressed, making TE a standard of care and maximally beneficial for liver disease sufferers worldwide. Further cost-effectiveness analysis comparing Liverscan against conventional TE and common serum biomarkers is required before its uptake as a point-of-care screening method.

**Supplementing TE with B-mode guidance.** Unlike conventional TE using a single-element transducer [9], Liverscan is configured with a phased transducer array for simultaneous sector B-mode image acquisition and LSM. Traditionally, operators solely relied on the difficult-to-interpret M-mode for liver localization. The incorporation of real-time B-mode imaging guidance provides multiple advantages over the current M-mode setting. First, B-mode possesses 2D spatial information, making it easier to differentiate the liver from the right kidney, gallbladder, or artifacts from nearby cardiac movement and lung or intestine gas. This eases the procedure of liver localization, thereby improving examination efficiency. The literature recognizes extreme BMI, higher liver stiffness, and chronic pulmonary disease as the risk factors of TE failure [37,41]; yet these linkages are poorly understood. One similarity in these conditions is the liver considerably deviates from its commonly located region, making the acoustic window challenging to search for.

Therefore, we postulate that B-mode guidance is of particular value to localizing the liver in those patients with post-partial hepatectomy, cirrhosis (shrunken liver size), emphysema (hyperinflated lung size), and extreme obesity. Second, the presence of intra-hepatic blood vessels, bile ducts, and focal masses could distort shear wave propagation trajectory [14] and pulsations around large vasculatures could lead to erroneously elevated shear wave speed [21]. Real-time guidance allows monitoring what intra-hepatic tissues are being measured without the inclusion of non-parenchymal components. This helps ensure the accuracy of LSM results. Third, the ability to capture anatomic images enables the recording of the exact location or landmark in the liver from which the LSM is made. This facilitates longitudinal monitoring and procedural standardization. Furthermore, Castera et al. [37] reported that limited operator experience (i.e., operation of <500 examinations) is the main determinant of LSM failure or unreliable results, probably due to the lack of B-mode guidance. Direct visualization of liver anatomy may contribute to shortening the learning curve, particularly for those without prior TE experience. Finally, beyond LSM, additional B-mode imaging could serve as a stand-alone modality for routine sonographic assessment of fatty liver and focal lesions (e.g., cyst, hemangioma, or hepatocellular carcinoma) [21].

In contrast, the phased array design presented in our study increases the geometry of the transducer face compared to the conventional single-element design. Although we have demonstrated the feasibility of using a larger transducer face for TE, future transducer design should consider dimensional specification to provide a compromise between a wider FOV and a higher risk of rib interference. Physical artifacts caused by diffraction and coupling of compression and shear wave [42–44] are another consideration, as they can lead to complex elastogram patterns that overestimate stiffness results. Sandrin's team [42] shed light on the relationship between circular vibrator diameter (from 1 to 20 mm) and the extent of these artifacts. The rectangular piston used in Liverscan does not necessarily create a point-like source of an elastic field, as the conventional circular vibrator does. Hence, the shape and size of a vibrating source should be carefully devised to minimize both diffraction and coupling effects. In practice, operators should be aware of the difference between the examination procedures performed by conventional TE and Liverscan (e.g., larger size of the transducer face and the addition of B-mode). Adequate training is necessary to determine optimal probe placement conditions to avoid ribs and large vessels.

Previous studies explored the incorporation of B-mode guidance into TE, but they have some pitfalls. Azar et al. [45] affixed a vibration actuator on the side surface of a conventional convex probe. Although this design offers a wider FOV, its curvilinear shape is subject to increased rib interferences due to a larger footprint and may lead to complex vibration sources. Another method involved physically coupling a TE probe with a conventional B-mode system equipped with a default convex probe [34,46]. However, this dual-probe setup required probe swapping during the procedure, and thus limited imaging guidance to a non-real-time basis. Our study was the first to report a fully integrated probe utilizing real-time sector B-mode as procedural guidance.

**Limitations and prospects.** We acknowledge the following limitations in this study. First, our study lacked histological data which serves as gold standard for liver fibrosis diagnosis. However, given the extensive validation and established status of Fibroscan as the leading non-invasive test of reference in the liver elastography field [21], the choice not to compare against histology can be justified [14,47,48]. Future studies should focus on establishing cutoffs for different fibrosis stages in biopsy-proven CLDs. Second, our cohort included patients with various liver diseases which may limit the generalizability of the finding to a specific etiology. Moreover, there might be a spectrum bias, because MASLD and CHB carriers accounted for 96% of the subjects in this study. Larger studies that adjust for important demographic variables (e.g., etiology, morphotype, and ethnicity) are necessary to further validate the performance of Liverscan. This will be critical for the roll-out and implementation of this technique in a range of clinical settings. Third, we did not specifically examine the added value of B-mode guidance to a TE examination procedure. This aspect had been in part addressed in a prior study by Lee et al. [34].

They confirmed the effectiveness of B-mode addition to mitigate TE failure, notably in a non-real-time format. Rather, our focus was on introducing an integrated probe that allows B-mode visualization of the liver while measuring its stiffness. Nevertheless, a thorough comparison of success rate and time consumption should be made among TE examinations with (a) real-time guidance, (b) non-real-time guidance, and (c) no guidance. Additionally, the applicability of the current probe setting to the pediatric population or adults of small stature remains unknown. The Fibroscan M probe has been reported to interfere with patients with narrow intercostal spaces, causing TE failure, unreliable results, and liver stiffness overestimation [49]. Since B-mode guided TE typically operates with a larger probe tip, rib interference due to changing anatomical relationships needs further verification.

## 5. Conclusions

A new ultrasound tool named 'Liverscan' has been developed to assess liver fibrosis at the point of care, which incorporates a wireless phased array probe to allow for B-mode guided TE in real-time. The system was evaluated in a laboratory setting with tissue-mimicking elasticity phantoms (2 kPa to 75 kPa) and in a clinical setting with Chinese CLD adults, showing sufficient accuracy, operator reliability, and comparable performance against conventional TE and 2D-SWE. Liverscan offers several technical advancements over existing modalities, such as real-time B-mode guidance for improved examination efficiency and a small footprint for mass screening and other point-of-care applications. Future research should involve more patients stratified by different etiologies and investigate the diagnostic accuracy in staging liver fibrosis using biopsy as the reference standard.

## 6. Patents

The patent entitled 'Method and Apparatus for Ultrasound Imaging and Elasticity Measurement' (US8147410 B2, CN101843501 B) was granted, resulting from the work reported in this manuscript.

**Supplementary Materials:** The following supporting information can be downloaded at: https://www.mdpi.com/article/10.3390/diagnostics14020189/s1, Figure S1: Bland–Altman plots of liver stiffness for (A) Conventional TE-fs vs. Liverscan; (B) Conventional-fs vs. conventional TE-ft; (C) Conventional TE-fs vs. 2D-SWE; Figure S2: Scatterplots illustrating linearity and correlation of liver stiffness between (A) Conventional TE-fs vs. Liverscan; (B) Conventional TE-fs vs. conventional-ft; (C) Conventional TE-fs vs. 2D-SWE.

**Author Contributions:** Conceptualization, Z.-H.H., L.-K.W., Y.Z. and Y.-P.Z.; methodology, Z.-H.H., L.-K.W., S.-Y.C., H.-X.C., Y.Z. and Y.-P.Z.; validation, Z.-H.H., L.-K.W., S.-Y.C., L.-K.C., Y.-W.L. and Y.-P.Z.; formal analysis, Z.-H.H., H.-X.C., M.-H.Z. and Y.-P.Z.; investigation, Z.-H.H., L.-K.C. and Y.-W.L.; resources, M.-H.Z. and Y.-P.Z.; data curation, Z.-H.H., S.-Y.C., H.-X.C. and Y.-W.L.; writing—original draft preparation, Z.-H.H. and Y.-P.Z.; writing—review and editing, Z.-H.H., Y.Z., L.-K.C., M.-H.Z. and Y.-P.Z.; supervision, Y.Z., M.-H.Z. and Y.-P.Z.; project administration, Z.-H.H., L.-K.C. and Y.Z.; funding acquisition, Y.-P.Z. All authors have read and agreed to the published version of the manuscript.

**Funding:** This research was partially funded by the Tai Hung Fai Charitable Foundation for Henry G. Leong, Professor in Biomedical Engineering, grant number 847 L.

**Institutional Review Board Statement:** This study was conducted in accordance with the Declaration of Helsinki, and approved by the Institutional Review Board of The Hong Kong Polytechnic University (application number: HSEARS20210809002; date of approval: 9 August 2021).

**Informed Consent Statement:** Informed consent was obtained from all subjects involved in the study.

**Data Availability Statement:** The data presented in this study are available on request from the corresponding author.

**Acknowledgments:** The authors acknowledge Miao-Qin Deng for her contribution to the algorithm development and Yoyo Lau for her assistance in the human clinical experiment, including patient enrolment, data acquisition, and data compilation.

**Conflicts of Interest:** Yong-Ping Zheng: Prof. Yong-Ping Zheng is the inventor of the patents related to the technique of B-mode imaging guided transient elastography (CN101843501 B, US8147410 B2, owned by The Hong Kong Polytechnic University) that have been licensed to the company Eieling Technology Limited, in which he is a co-founder, shareholder, and director. Li-Ke Wang: Mr. Li-Ke Wang is a full-time employee of Eieling Technology Limited. The other authors declare no competing interests.

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
