# Peer review of "Palm-Sized Wireless Transient Elastography System with Real-Time B-Mode Ultrasound Imaging Guidance: Toward Point-of-Care Liver Fibrosis Assessment"

_diagnostics, doi:10.3390/diagnostics14020189_

Round 1

Reviewer 1 Report

Comments and Suggestions for Authors

Very good and interesting paper! The issues:
1. The term NAFLD was replaced by MASLD in 2023 (Rinella); use this term.
2. The conclusions are rather lenghty; two or three sentences should suffice. 

Reviewer 2 Report

Comments and Suggestions for Authors

Zi-Hao Huang with coauthors wrote an article to introduce Liverscan® and compare its results to other well-known liver elastography assessing devices.

The importance of this study is high as the Authors point out some disadvantages of classical transient elastography (TE) and present a device that fills the missing gap in TE.

From a technical point of view, this manuscript is written clearly, and the paragraphs are consistent with their titles. I noticed the size of the full text while viewing it for the first time. The text is very vast. When I had read it all, I understood that the Authors wished to explain every step of their work, from preparations to tests on patients.

Every issue I was wondering about while reading was explained in the text, thus I have nothing to point out to correct or add.

The topic was also interesting for me because I am currently in a group of physicians that compares similar device as the Authors. It is not laptop/tablet size, it is wired, but also combines elastography with B-mode ultrasound allowing it to find a place in the liver free of blood vessels or large bile ducts. Therefore, I am aware of how the B-mode ultrasound can improve the examination. After collecting a little over 100 examinations with Fibroscan® and the tested device, we have analyzed the results achieved by these two devices and – I am writing it with a little dose of astonishment – we got very similar results with a bit higher elasticity in tested device against the Fibroscan® which was also around of 0.8 kPa. I do not know what the beam width is in the tested device thus I cannot tell you that this is the reason for this observation. Nevertheless, both the Authors and our study team got the same differences between Fibroscan® and the tested device.

Good luck, I will look forward to the next studies, especially in setting the elasticity thresholds for different stages of fibrosis in the METAVIR scale.
